# Macroscopic waves, biological clocks and morphogenesis driven by light in a giant unicellular green alga

Eldad Afik [1,2] ✉, Toni J. B. Liu [1] & Elliot M. Meyerowitz [1,2] ✉

A hallmark of self-organisation in living systems is their capacity to stabilise their own dynamics, often appearing to anticipate and act upon potential outcomes. *Caulerpa brachypus* is a marine green alga consisting of differentiated organs resembling leaves, stems and roots. While an individual can exceed a metre in size, it is a single multinucleated giant cell. Thus *Caulerpa* presents the mystery of morphogenesis on macroscopic scales in the absence of cellularization. The experiments reported here reveal self-organised waves of greenness − chloroplasts − that propagate throughout the alga in anticipation of the day-night light cycle. Using dynamical systems analysis we show that these waves are coupled to a self-sustained oscillator, and demonstrate their entrainment to light. Under constant conditions light intensity affects the natural period and drives transition to temporal disorder. Moreover, we find distinct morphologies depending on light temporal patterns, suggesting waves of chlorophyll could link biological oscillators to metabolism and morphogenesis in this giant single-celled organism.

A free-falling ball follows a trajectory determined by external forces. When we aim to catch the ball, in contrast, our dynamics follows an intrinsic predictive model of the world, exhibiting anticipatory behaviour[1–3]. Such behaviour has been observed in living systems down to cellular levels[4–7].

Time-keeping and synchronisation with the external world, as well as within the organism, seem to be essential for homoeostasis, the ability of living systems to maintain essential variables within physiological limits[1,8–10]. This is one of the ways by which living organisms manifest self-organisation[11], defined here as networks of processes that are self-stabilizing far from thermodynamic equilibrium.

Biological oscillators are found throughout the living world, constituting a mechanism for organismal synchronisation. Their manifestations, such as pulse rate and pressure, are measures of homoeostasis. Biological oscillators are amenable to analytic approaches from dynamical systems[9,12,13]. Here we study active rhythmic transport in *Caulerpa brachypus*, a marine green alga which presents complex morphology while being a giant single cell. *Caulerpa* challenges central paradigms in developmental biology, as it exhibits

pattern formation and morphogenesis without multicellularity[14,15]. While tracking morphogenesis of regenerating algal segments we observed that the growing tips of stem-like (stolon) and leaf-like (frond) regions exhibit temporal variations in intensity of their green colour. During night distal regions typically appear more transparent, while during day the green coverage is more homogeneous. This has been attributed to long-distance chloroplast migration[16]. Whether rhythmic transport in this single cell is light induced, or is of autonomous nature, is unknown. Our findings show that the waves of greenness exhibit anticipatory behaviour, usually starting toward the daylight state before dawn, and toward the night state before dark. Moreover, the analysis shows that anticipatory behaviour in this system is explained by entrainment—adjustment of the rhythm of an organismal self-sustained oscillator by interaction with another oscillator[17,18], the driving illumination in this case. Using a scalable and affordable methodology, varying two control parameters—the temporal period of the illumination and its intensity—we identify distinct dynamical states of the self-organised waves. The results are compatible with dynamics of self-oscillations and forced

[1]Division of Biology and Biological Engineering, California Institute of Technology, 1200 E California Blvd., Pasadena, CA 91125, USA. [2]Howard Hughes Medical Institute, Maryland, USA. ✉e-mail: eldad.afik@gmail.com; meyerow@caltech.edu

synchronisation[18]. In constant illumination, the intrinsic period is intensity dependent, and under higher illumination intensities analysis indicates a transition to states of increased temporal disorder. Moreover, we find that development depends not only on the average photon flux but also on its temporal distribution, as manifested in the resulting morphology.

## Results

### Anticipatory green waves and morphogenesis

To study quantitatively the self-organised green wave dynamics in *Caulerpa brachypus*, we developed a set of protocols for algal culture and propagation, as well as automated illumination, live imaging and image analysis. Using Raspberry Pi-controlled illumination and cameras, we track over weeks the morphogenesis of tens of samples concurrently, while tracing at resolution of tens of seconds the variation of the green coverage. A diagram of the experimental flow is shown in Fig. 1a; snapshots exemplifying the relevant time scales are presented in Fig. 1b–d. Our observations of samples cultured under cycles of 12 hours light followed by 12 hours dark (12hL–12hD, T = 24h) have allowed analysis of waves of greenness that propagate over centimetres, within a few hours, at a whole-organism scale; a time-lapse is presented in Supplementary Movie 1; *Light* and *Dark* denote reference illumination intensities, the latter is $1/200 \cdot I_L$, for $I_L \approx 4.5 \mu mol \cdot s^{-1} \cdot m^{-2}$. By coarse-graining in space we achieve a reduced description to a dynamic macroscopic observable. The time series indicates that the initiation of the waves anticipates the external change in illumination, as exemplified in Fig. 1e. This has led

us to ask whether the period of the waves follows that of the driving illumination.

### Equivalence of dynamics near 24h driving periods

To quantify the temporal frequency content of the biological response, we apply power spectral analysis[19,20]. While the time series of individual samples exhibit high variability, the spectral decomposition using power spectra captures and highlights the similarities among them. Examples from three samples and their corresponding power spectra are plotted in Fig. 2a and b, respectively. The pronounced local maxima in the power spectra correspond to the response fundamental frequency $f_{r,0}$, centred at 1/24h for these samples, and its higher harmonics—integer multiples of $f_{r,0}$. The relative power of the higher harmonics characterises the non-sinusoidal waveforms shown in Fig. 2a.

Under driving periods within 18h to 30h the response of the green pulses $f_{r,0}$ matches the driving frequency $f_d$ in 1:1 correspondence, as shown in Fig. 2d. Moreover, the power spectra suggest an equivalence of the dynamics when rescaled by the control parameter $f_d$. That is, presenting the power spectra from various $f_d$ as a function of $f_r/f_d$ reveals the similarity in dynamics in this range; see Fig. 2e. The corresponding curves prior to rescaling are shown in Fig. 2c.

### The waves are coupled to an intrinsic oscillator

Subject to driving periods away from the 18h to 30h range, we find in addition to the driving frequency evidence for a distinct circadian mode—an indication that the waves are coupled to, or part of, an

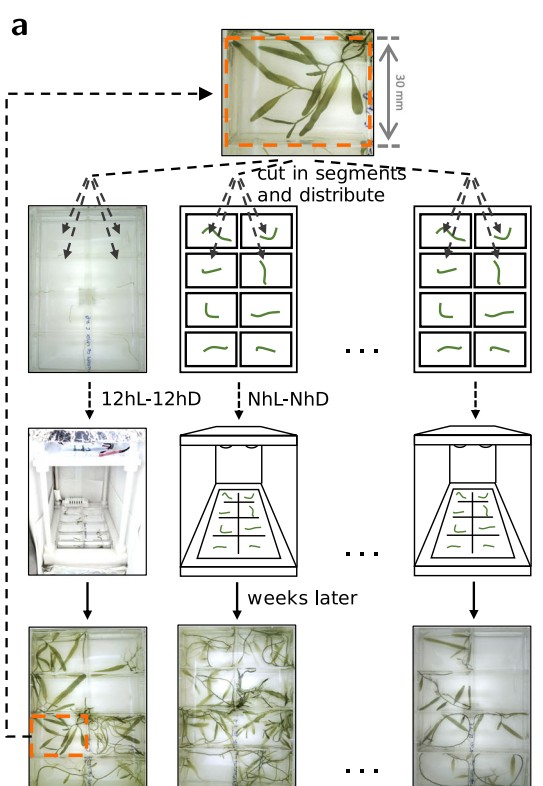

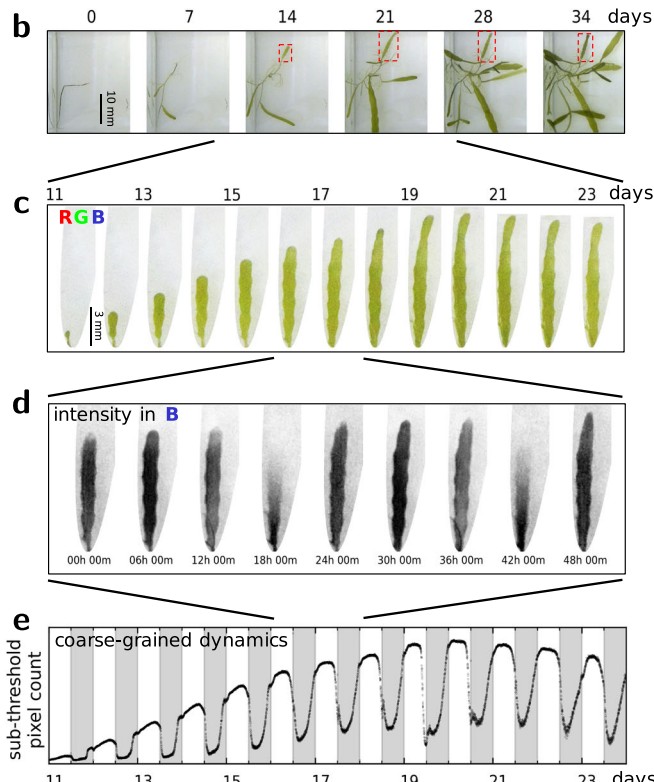

**Fig. 1 | Self-organised macroscopic green waves: experimental design and coarse-graining. a** Schematic of the Cut & Regenerate experimental cycle. A *Caulerpa* cell is cut and segments are let to regenerate in individual wells; custom-built controllers apply the individually assigned illumination protocols per dish and control the time-lapse imaging; samples regenerating over weeks provide tens of segments for the next experiment; thus the experiments provide living material for the next cycle of experiments. **b** Snapshots from morphogenesis dynamics over the course of 35 days. **c** A developmental trajectory of a frond, focusing on a sub-

interval of 13 days; these correspond to the frond marked by red rectangles in (**b**). **d** A temporal sub-interval spanning 48h, showing two periods of the green waves; the grey-scale frames represent the intensity in the Blue channel from the RGB images. **e** Coarse-graining of the spatio-temporal dynamics achieved by counting pixels whose intensity in the Blue channel falls below a preset threshold; the white and grey shadings correspond to 12h intervals of Light and Dark illumination levels, respectively.

  

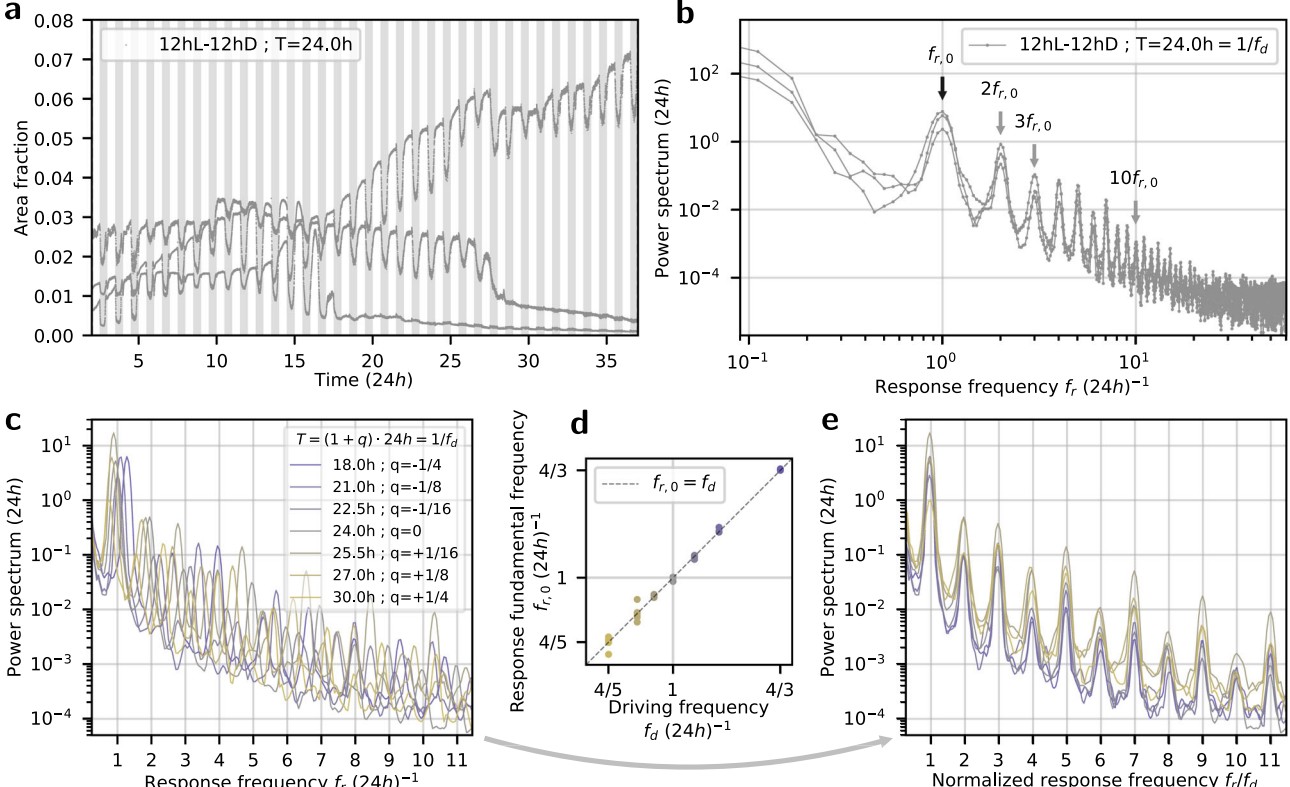

**Fig. 2 | Dynamical equivalence under driving periods near 24h. a** Time series from three samples regenerating under 12hL-12hD (T = 24h); area fraction corresponds to the measured green area within a whole well, normalised by the area of the well. **b** Power spectra corresponding to the time series in (**a**); the arrows annotate the 1st, 2nd, 3rd and 10th local maxima, corresponding to the response fundamental frequency $f_{r,0}$ and higher harmonics, namely integer multiples of $f_{r,0}$. **c** Power spectra of samples regenerating under illumination periods within 18h to 30h; the curves result from averaging over samples grouped by driving illumination frequencies, $f_d = 1/T$. **d** Response fundamental frequencies $f_{r,0}$ plotted as function of the driving frequency $f_d = 1/T$; the guiding dashed line represents a 1:1 correspondence. **e** Power spectra presented as a function of $f_r/f_d$, the response frequency rescaled by the driving frequency. The power spectra whose averages are shown in (**c**) and whose $f_{r,0}$ are in (**d**) can be found in Supplementary Fig. 1.

intrinsic nonlinear oscillator. For example, under driving period T = 48 h time series exhibit a steady phase relation to the illumination pattern; at the same time, at mid-Light intervals the dynamics resemble the day to night transitions observed under T = 24h; see Fig. 3a. Under a driving period T = 3 h, eight times faster than earth's rotation, the time series show fast undulations attributed to a response to the driving illumination; these ride over an undulation of a longer period, as shown in Fig. 3b.

Entrainment of a nonlinear oscillator is predicted to occur at driving frequencies near rational ratios of its natural frequency and the driving one, in which case *n* cycles of the response would match *m* cycles of the driving oscillator[17,18]. Within an *n:m*-entrainment range, the response fundamental frequency $f_{r,0}$ and its higher harmonics are predicted to be modified accordingly. To test for entrainment, we have measured samples subject to $f_d$ about rational ratios of 24h, namely T = 48 h, 6 h, 3 h, and 1.5 h, corresponding to $2^1, 2^{-1}, 2^{-3}$ and $2^{-4}$ multiples of 24h, as well as longer ones T = 94 h and 54 h. The results support a 1:1-entrainment range, as well as evidence for higher order *n:m*-entrainment regions. The corresponding power spectra are presented as heatmaps, where the columns correspond to $f_d$, including those from $f_d$ near 1/24h; see Fig. 3c, a visualisation inspired by a study on forced turbulence and its synchronisation regions[21]. The synchronisation regions, where *n:m*-entrainment are stable, are known as Arnold tongues[17,18,21].

### Dynamical states set by the intensity of light

Finally, we test whether alternating environmental conditions are necessary to sustain the waves, to complete the argument for an intrinsic self-sustained oscillator, and measure its natural period. To this end, we let samples regenerate under constant illumination. While we indeed find self-sustained circadian oscillations, subject to constant illumination the observations indicate new dynamical states. Examples of time series regenerating under low driving intensity $I_d = 2^{-5} \cdot I_L$ show a reduced amplitude of the oscillations, yet these persist over weeks; see Fig. 4a. Samples subject to high driving intensity $I_d = 2 \cdot I_L$ show signs of intermittent oscillations, presented in Fig. 4b. Power spectral analysis shows that: (i) the response fundamental frequency $f_{r,0}$ increases with decreasing illumination intensity, see Fig. 4c; (ii) under higher illumination intensities the local maxima in the power spectra 'drown' in a rising continuum of frequencies, and (iii) compared with the entrained states, the higher harmonic content is less pronounced; these findings are presented in Fig. 4d. Thus, under constant photon flux, the energy source for this organism, we find that the system can be driven towards aperiodic dynamical states by high fluxes. This is a manifestation of the nonlinear nature of the dynamics, and a plausible signature of chaotic dynamics.

### Morphology depends on the temporal pattern of light

How does the photon flux affect the developmental dynamics in this organism? It is expected that the energy injection rate would be a limiting factor to growth. Inferring the apparent area of samples from our macroscopic dynamical variable, we find that both the central tendency and the dispersion of growth rates increase with the illumination intensity; that is, algal samples regenerating under high photon flux cover larger areas faster as a population, while being less predictable individually, compared with those regenerating under a low

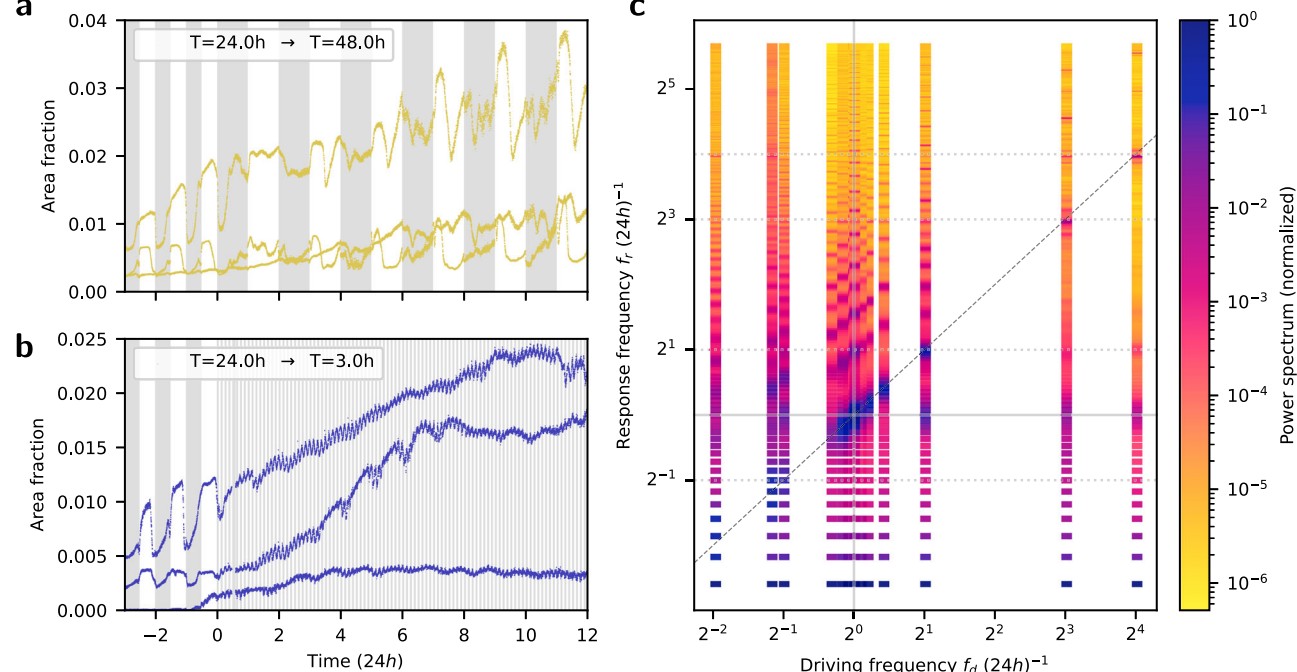

**Fig. 3 | The waves are coupled to, or part of, an intrinsic nonlinear oscillator, in addition to the driving illumination for periods far from 24h. a** Time-series from three samples regenerating under driving illumination of T = 24 h (12hL-12hD) switching to T = 48h (24hL-24hD); while each is distinct, the curves reveal a steady phase relation to the illumination pattern. **b** Time series from three samples regenerating under T = 24 h (12hL-12hD) switching to T = 3h (1.5hL-1.5hD); note that the fast undulations, which follow the driving illumination, are riding on a longer period one, close to 24h. **c** Power spectra presented as heatmaps; each column results from averaging over samples, grouped by driving illumination frequencies, $f_d = 1/T$; the guiding dashed line represents a 1:1 correspondence; the columns near $f_d$ of 1/24h overlap with those in Fig. 2c, presented here to demonstrate 1:1-entrainment range; away from this region, in addition to the spectral content associated with the driving frequency $f_d$, local maxima are found near 1/24h (grey horizontal line), an indication of an intrinsic oscillator of circadian frequency. The power spectra whose averages are shown in (**c**) can be found in Supplementary Figs. 1 to 3.

photon flux. Moving averages, as estimates of apparent sample area, are presented in Fig. 4e. However, it turns out that the temporal pattern of the illumination leads to distinguishable morphological traits, even when the mean photon flux is kept the same. For example, we compare two mean photon flux levels, $2^{-1} \cdot \mathtt{I_L}$ and $\mathtt{I_L}$, each at constant illumination and T = 24h, presented in Fig. 5. The comparison hints that the temporal distribution of photons, analogous to time-restricted feeding, impacts development as a control parameter which is independent of the mean photon flux.

## Discussion

This study reveals self-organised macroscopic waves in a unicellular organism. Using coarse-graining analysis and experimental perturbations we have identified distinct dynamical states of the waves. This report provides evidence for coupling to an intrinsic self-sustained oscillator, which is entrained by the time-dependent driving illumination, thus explaining the anticipatory behaviour of the waves. Under constant conditions we find that the natural frequency depends on the photon flux, and that the dynamics exhibits a transition to temporal disorder. Furthermore, our findings of distinct morphologies tie the discovered waves to one of the mysteries of development in macroscopic single cells, morphogenesis.

*Caulerpa* feeds on photons, so metabolism is likely to be a key to understanding the observed entertainment by light, via photosynthesis. This is the case in the KaiABC system—a molecular circadian oscillator of certain cyanobacteria[22,23]. Moreover, considering photosynthesis sheds new light on our identification of distinct dynamical states, subject to the two control parameters—illumination intensity and period. Constant illumination by itself does not drive the system to temporal disorder. Two of the driving protocols we have studied here are equivalent in their average photon flux, namely 12hL-12hD and

constant $\mathtt{I_d} = 2^{-1} \cdot \mathtt{I_L}$. The latter leads to increased temporal disorder dynamics, hinting that relaxation in the dark is essential for the typical organismal dynamics.

How do tissues and organs emerge on centimetre scales in the absence of cellularization? Active transport has been hypothesised to play a key role in *Caulerpa* regeneration[14,15]. The observation of the self-organised waves leads us to postulate that their synchronisation with light-driven metabolic switching may hold the answer. This hypothesis assumes two key elements: (i) spatial distribution of chloroplasts would result in local variation in light-harvesting efficiency, and (ii) light-dependent metabolic states of chloroplasts. Under these assumptions, redistribution of chloroplasts in anticipation of the temporal changes in photon flux could create and stabilize metabolic sub-environments, akin to localised organs. An additional contribution of the waves to the dynamics may come in the form of an effective pump: cortical mass migration of chloroplasts could result in bulk flow, which in turn drives cytoplasmic streaming; dynamics of this kind has been demonstrated in other macroscopic cells[24].

Ashby[11] has proposed a paradigm for self-organisation, where a subset of variables in a dynamical system can undergo abrupt transitions between two values. Such transitions would result in dynamical switching among distinct behaviours of the whole system. In this light we propose that biological oscillators, morphogenesis, and metabolism are interconnected sub-systems within a network of processes; their dynamics and inter-relations lead to the emergence of self-stabilisation far from thermodynamic equilibrium in this natural system which is alive—a generalised homoeostasis.

## Methods

To quantitatively study the green wave dynamics in *Caulerpa brachypus*, we have developed an affordable and scalable experimental

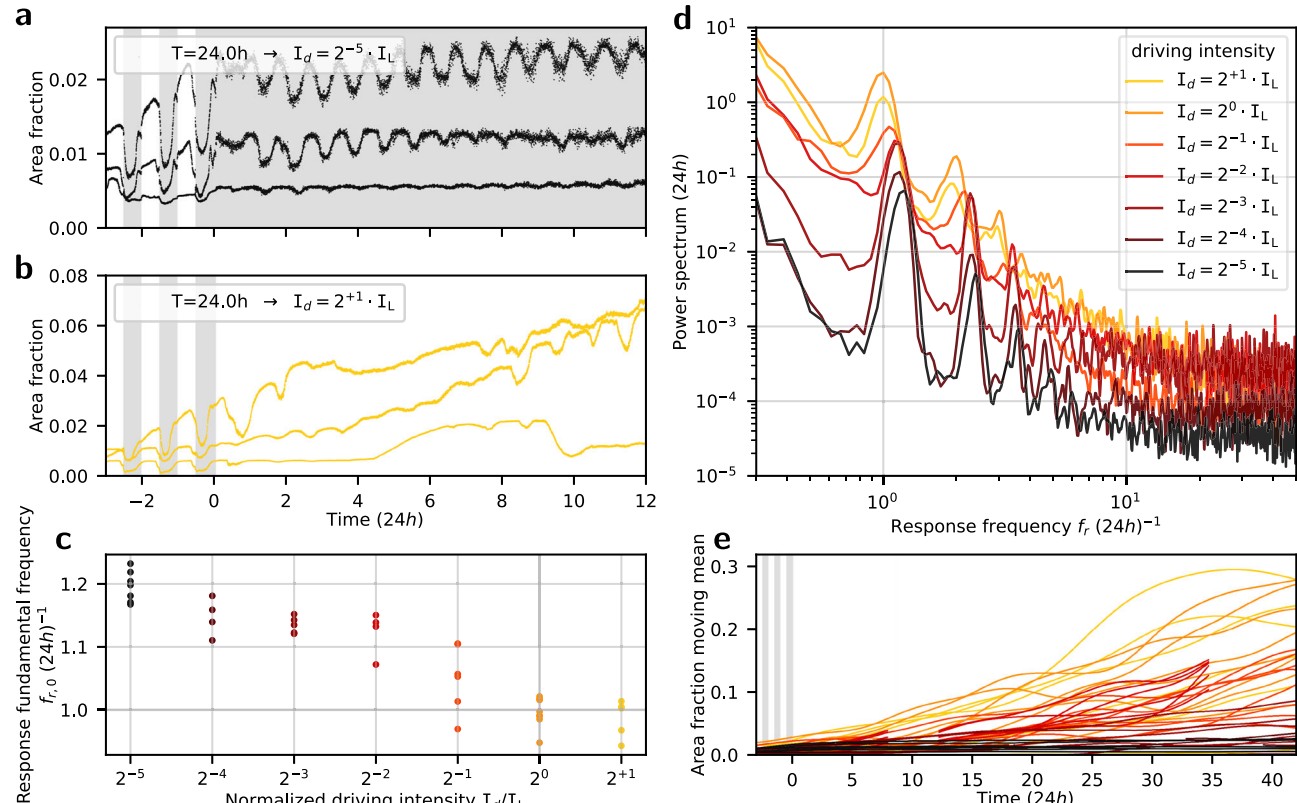

**Fig. 4 | Free-running at constant-in-time illumination reveals intensity dependent dynamical states. a** Time series from three samples regenerating under driving illumination of T = 24h switching to constant intensity $I_d = 2^{-5} \cdot I_L$. **b** Time-series from three samples regenerating under driving illumination of T = 24h switching to constant-in-time intensity $I_d = 2 \cdot I_L$. **c** Response fundamental frequencies $f_{r,0}$ plotted as function of the driving intensity, $I_d$; the data indicate an intensity dependence of the response fundamental frequency when the conditions are constant-in-time. **d** Power spectra inferred from samples regenerating under

driving illumination intensities within $2^{-5} \cdot I_L$ to $2^{+1} \cdot I_L$; the curves result from averaging over samples grouped by driving illumination intensities, $I_d$; under higher driving intensities the data reveal an increase in power distribution across a continuum of frequencies. **e** Moving averages of the time-series for samples regenerating under constant illumination; in addition to intensity-dependent growth rates, the data reveal a sample dispersion which increases with the driving intensity. The power spectra whose $f_{r,0}$ are shown in (**c**) and whose averages are in (**d**) can be found in Supplementary Fig. 4.

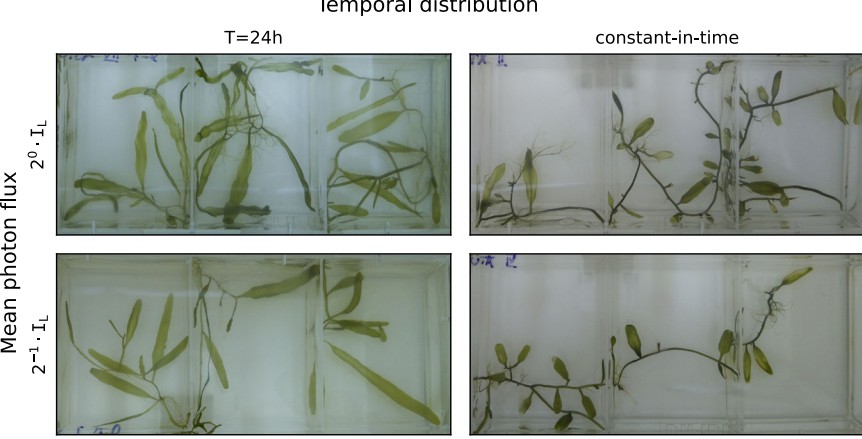

**Fig. 5 | Developmental effect of photon flux intensity and temporal distribution.** Snapshots from four driving illumination conditions, comparing mean photon flux levels $2^0 \cdot I_L$ and $2^{-1} \cdot I_L$, at T = 24h and constant illumination. Samples regenerating under time dependent illumination at T = 24h share qualitative morphological features, which are distinguishable from those under constant illumination, for example, the typical frond size and aspect ratio. The panels are

cropped to show three wells out of eight from each culture dish; following 13 days subject to T = 24h, $2^{-1} \cdot I_L$ (12hL-12hD) the new driving illumination was applied; panels correspond to 25 days after this transition. The developmental effect distinguishing between the temporal distributions compared above−T = 24h and constant illumination−are not as apparent in growth curves; corresponding moving averages of the time series can be found in Supplementary Fig. 5.

system for the algal culture, propagation and imaging, summarised in Fig. 1a. A Raspberry-Pi-Zero-W is mounted on a custom designed 3d-printed apparatus, such that its camera module is focused at an 8-well-culture-dish. Using multiple replicas, we concurrently track over weeks the morphogenesis of tens of samples, while tracing at resolution of tens of seconds the variation of the green coverage. The same computers additionally control LED strips, assigning distinct time-dependent illumination per culture dish that provides both for photosynthesis and imaging. Samples regenerating over weeks can provide tens of segments for the next experiment; hence the experiments provide living material for their follow-ups in a cyclic manner. A time-lapse of samples regenerating under 12hL-12hD, 24h Light-Dark cycles, is presented in Supplementary Movie 1.

Towards reproducibility, controlling for the variability in initial conditions of the living samples, we use the regenerative properties of *Caulerpa brachypus*. Individual algal samples are cut in segments of 1–3 centimetres in length; within 1–2 months a regenerating sample can provide for about 20 new segments. This approach contributes to the homogeneity of the observed individuals, both in their genetic background, as well as the developmental state and geometry set as initial conditions. A visual summary of the experimental setup is presented in Fig. 1a.

The nature of the data presents both conceptual and computational challenges. Conceptually it is desired to identify a coarse-graining procedure, to reduce the high-dimensionality of the spatio-temporal dynamics of the green waves, and at the same time preserve a minimal set of essential degrees of freedom. Images are taken every 150 s over weeks. This amounts to datasets of about 400 GiB/month/dish of RGB (Red-Green-Blue) images. Running several experiments concurrently leads to high-throughput data generation. It is therefore challenging for human detailed inspection, introducing computational challenges: for computer vision and for computational resources.

In this report, our dynamic observable is the apparent green area. Its time series are inferred from applying a threshold to the blue channel of the RGB images. Chlorophyll absorbs strongly in the blue. Therefore regions rich in Chlorophyll are expected to show relative darker values in the Blue channel. An example is summarised in Fig. 1b–e, where the time series correspond to a region of interest tracking a single leaf-like organ. Everywhere else in this report the time series have been inferred from regions of interest corresponding to wells; hence these represent the apparent fraction of green area, where the area of the well is the normalisation factor.

### *Caulerpa brachypus* culture and propagation

*Caulerpa brachypus* algal samples in this report were all derived from a single origin (shipped from Inland Aquatics, Indiana).

As culture medium we use ErdsSchreiber's Medium (UTEX Culture Collection of Algae at The University of Texas at Austin). It is a variation over the one in previous reports on *Caulerpa*[25,26].

Prior to cutting, samples were triple washed using filtered artificial sea water (35% Instant Ocean in deionized water; filtered using a Nalgene™ Rapid-Flow™ with 0.1 $\mu$m PES Membrane filters, Thermo Fisher Scientific Inc.). Algal segments were cut using a surgical blade, and let to regenerate in 8-well-dishes (Non-treated Nunc™ Rectangular Dishes, Thermo Fisher Scientific Inc.), each well filled 7ml of the culture medium.

### Illumination and imaging setup

Custom made apparati were 3D-printed using translucent neutral colour Polylactic Acid (PLA and 3D Printers access courtesy of Caltech Library Techlab). These were designed to hold the culture dish, as well as the LEDs, Raspberry-Pi and camera module.

Raspberry-Pi-Zero-W and its 8MP Camera v2 module (Adafruit Industries) were used for time-lapse image acquisition (using the RPi-Cam-Web-Interface). Exposure time and gain were automatically set by

the software to allow consistent imaging at various illumination intensities. The Raspberry-Pi also controlled the illumination protocols, allowing the assignment of time-dependent photon-flux per culture dish. This was done by pulse-width modulation (using the Python interface of the pigpio library) of LED strips (5000K cool-white SMD5050 12V, L1012V-502-1630 from HitLights), two triplets per dish. Reference illumination intensity is estimated at $I_L \approx 4.5 \, \mu mol \cdot s^{-1} \cdot m^{-2}$, measured at the culture medium level covered by the culture dish lid (photodiode power sensor connected to an optical power metre, S120C and PM100D from Thorlabs). Under these conditions, the measured illumination spectrum shows two main modes, one at 545nm, half-max extends 505nm-614nm, and the other at 440nm, half-max extends 428nm-449nm (optic spectrometer USB2000+ from OceanOptics).

The experiments were conducted in incubators (DT2-MP-47L from Tritech Research, Inc., and VWR 2015 from Sheldon Manufacturing Inc.) at temperatures maintained within 22.5°C to 24.5°C.

### Spatial coarse-graining, post-processing and inference of power spectra

Images have been split into regions of interest, each corresponding to a well in the 8-well-dish. For each region of interest, time series correspond to the fraction of pixels whose value does not exceed a threshold of 70 out of 255 in the blue channel.

Post-processing of the time series consists of outlier detection and replacement. Outliers have been defined as those exceeding 75% of the well, or lying outside the 1.5 × Interquartile-Range of a rolling window spanning 1h40m. Where there are no more than three consecutive outliers, these have been replaced by linear interpolation.

**Table 1 | Sample sizes per illumination protocol organised by the figure to which these contributed**

| Fig. 2 and Fig. 3 | |
| --- | --- |
| T | sample size |
| 18h | 5 |
| 21h | 5 |
| 22.5h | 7 |
| 24h | 3 |
| 25.5h | 8 |
| 27h | 8 |
| 30h | 5 |
| **Fig. 3** | |
| T | sample size |
| 1.5h | 6 |
| 3h | 7 |
| 12h | 6 |
| 24h | 27 |
| 48h | 8 |
| 54h | 4 |
| 94h | 5 |
| **Fig. 4** | |
| $I_d$ | sample size |
| $2^1$ | 6 |
| $2^0$ | 8 |
| $2^{-1}$ | 8 |
| $2^{-2}$ | 6 |
| $2^{-3}$ | 6 |
| $2^{-4}$ | 5 |
| $2^{-5}$ | 7 |

In Fig. 3c, the 27 samples subject to T = 24h are from 4 culture dishes, from 3 distinct experiments.

Power spectra have been inferred based on 35 day intervals, starting two days after transitioning from 12hL-12hD to the new driving illumination protocol. Following the above post-processing procedure for outlier detection and replacement, time series consisting of more than 2% unreplaced outliers within these intervals were excluded from this report.

Power spectra have been estimated by Welch's method[19], applying Hann taper to segments of 8.5 days (4896 time-points), 50% overlap between segments, zero-padded for interpolated frequency resolution of $1/18 \times (24h)^{-1}$, without detrending. Any remaining detected outliers were replaced by forward and backwards propagation of nearest valid data.

To estimate dominant frequencies, local maxima in the power spectra are detected by a two step process: first, local peaks are filtered by requiring a prominence of at least twice the level of their baseline; second, these are refined by a local fit to a parabola on logarithmic scale, applied to 5 points centred at the peaks detected by the first step.

The trends in Fig. 4e have been estimated by a rolling average, using gaussian weighting window of $\sigma$ corresponding to 2 days 00h55m45s, and spanning $4\sigma$.

**Sample size.** Experiments were designed to result in at least 3 samples per illumination protocol. Illumination protocols were applied to at least one culture dish, each consisting of 8 wells. Time-series exclusion procedure based on outlier detection is detailed above. The resulting sample sizes per illumination protocol are listed in Table 1, organised by the figure to which these contributed. A degree of sample dispersion within illumination protocols can be appreciated from Fig. 2d and Fig. 3c.

## Reporting summary

Further information on research design is available in the Nature Portfolio Reporting Summary linked to this article.

## Data availability

The datasets generated and analysed during the current study are available in the figshare repository, https://doi.org/10.6084/m9.figshare.23797020[27].

## Code availability

All programming and computer aided analysis has been done using open-source projects, primarily tools from the Scientific Python ecosystem[28]: SciPy[29], pandas[30], IPython and JupyterLab[31]; image processing has been distributed using dask and xarray; visualisation has been done using HoloViz and Matplotlib. Custom computer codes used to analyse the results reported in the manuscript are available in the figshare repository, 10.6084/m9.figshare.23797020[27], and from the corresponding authors on reasonable request.

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

## Acknowledgements

The authors thank A. Kornfeld for the computer-aided design of the 3d-printed apparatus, as well as Caltech Library Techlab and staff for 3D Printers access, materials and support. We are grateful to L.J. Schulman, M. Mussel and T. Li for stimulating discussions through various stages of the research, as well as to L. Michaeli, A.I. Flamholz, G. Manella and S.A. Wilson for helpful discussions and critical comments on the manuscript. E.A. is thankful for fruitful discussions at the SRBR2022 and EBRS2022 meetings. Distributed image processing was conducted in the Resnick High Performance Computing Center, a facility supported by Resnick Sustainability Institute at the California Institute of Technology. The laboratory of E.M.M. is supported by the Howard Hughes Medical Institute. E.A. has been awarded the Zuckerman Israeli Postdoctoral Scholar, Zuckerman STEM Leadership Program, and the Biology and Biological Engineering Divisional Fellowship, Caltech. T.J.B.L. has been awarded the Summer Undergraduate Research Fellowship (SURF), Caltech. This article is subject to HHMI's Open Access to Publications policy. HHMI lab heads have previously granted a nonexclusive CC BY 4.0 license to the public and a sublicensable license to HHMI in their research articles. Pursuant to those licenses, the author-accepted manuscript of this article can be made freely available under a CC BY 4.0 license immediately upon publication.

## Author contributions

Conceptualisation: E.M.M proposed studying morphogenesis in *Caulerpa*; E.A. designed the study; Methodology: E.A. designed the experimental system and analysis; Investigation: E.A. performed the measurements; E.A. and T.J.B.L. performed computational analysis; Visualisation: E.A. and T.J.B.L.; Funding acquisition: E.M.M.; Writing—original draft: E.A.; Writing—review & editing: E.A. and E.M.M.; All authors discussed and commented on the manuscript.

## Competing interests

The authors declare no competing interests.
