## [Peer Review File · Nature Communications]

Macroscopic waves, biological clocks and morphogenesis driven by light in a giant unicellular green algaREVIEWER COMMENTS

Reviewer #1 (Remarks to the Author):

This paper focuses on the understudied multinucleate alga of the genus *Caulerpa*, which undergoes morphogenesis despite being a single cell. The paper nicely demonstrates that the alga undergoes circadian rhythms in the localisation of chloroplasts, which migrate to the base of the cell during night. The migratory movements anticipate the arrival of light or dark and can persist under low constant light, demonstrating the action of a circadian oscillator that can be entrained. The paper further shows that growing the alga under a day-night cycle favours growth compared to growing under constant illumination with the same average light intensity, indicating that the circadian response of the organism contributes to its ability to grow and spread. All of these are valuable and novel findings that bring description and analysis of circadian rhythms to this understudied system.

However, the link between the presented findings and the mystery of unicellular morphogenesis, which as the authors point out is the most intriguing aspect of *Caulerpa*, is weak. The effect of continuous compared to cyclic illumination seems to be reduced growth, and while this may affect aspect ratio of the frond, this appears to be no more instructive for understanding morphogenesis than showing that the aspect ratio of a leaf can be altered by growing a plant under abnormal conditions that retard growth. Moreover, it is unclear whether the effect on growth is caused by the altered migration of chloroplasts or other metabolic consequences of continuous illumination.

Reviewer #2 (Remarks to the Author):

This work analyzes the dynamics of chloroplast waves in green algae and how these periodic dynamics entrain to forcing over time. The data and results in this work are quite profound, providing evidence of complex spatiotemporal dynamics and patterns over space and time. These results will have significant implications for the biology and applied math communities. The results are very well presented (excellent figures, very thorough analysis, clear writing). I am strongly in favor of the publication of this work.

One minor revision I would like to see is more detail on the dynamical systems modeling/analysis. Specifically the coupled oscillator model is mentioned, but I had a hard time finding the details in the paper to connect to the figures and results. Even just a quick summary or highlights would be nice. Not necessary though as I realize this is likely a whole set of follow up papers.

Reviewer #3 (Remarks to the Author):

The authors study an interesting example of spatiotemporal self-organization – the light driven growth of a giant unicellular green alga. Their study combines nonlinear dynamics, developmental biology, and chronobiology. Thus the results should be of interest to a broad range of readers. I find the enormous amount of quantitative imaging data impressive (compare Fig.1). A system was established to combine short term dynamics, daily rhythms, and long-term growth. The huge amount of processed data becomes visible in their comprehensive Figures 3 and 4.

The challenge was to condense all these data to a focused manuscript. Using global characteristics such as normalized areas and Fourier analysis many exciting results could be extracted: indications of self-sustained rhythms, entrainment to a wide range of light-dark cycles, and relevance of entrainment for morphogenesis.

In order to learn even more from these great data of an interesting model organism I list below some technical questions I would like to be discussed. Some answers might be also included into the revised manuscript which is already of high quality.

Specific comments and questions:

1. "... waves ... propagate over centimetres, within a few hours ..." This propagation seems really fast compared to normal diffusion. The authors might discuss some possible explanations: convection, active transport, anomalous diffusion, trigger waves (see e.g. J Ferrell 2018).

2. Figure 2 contains a lot of somewhat hidden information. I recommend providing the curves in Fig.2c individually as supplemental material. This could help to address some follow up questions: Is there a kind of resonance (amplitude effects)? Are there the expected relationships between driver period and entrainment phase?

3. It is claimed in Figure 3 that the intrinsic oscillations are "self-sustained"? Indeed, the rhythms in Fig.3b are persistent over many cycles. It has been shown, however, that a distinction between weakly damped oscillations and limit cycles is difficult even for recordings of more than 40 cycles (TL Leise 2012, D Gonze 2005, PO Westermark 2009). It was shown recently, that light-dark conditions can also drive transitions from self-sustained to damped oscillations (P Burt 2021). The wide range of entrainment (or resonances) in Fig.3c point to a weak intrinsic oscillator that can be entrained easily to a wide range of periods (as in *Neurospora crassa*, see M Merrow 1999).

4. Driving the system with different T-cycles (DJ Dijk 1999) reminds me to the "forced-desynchrony protocols". In these cases no entrainment is expected for the strong human clock. A comparison might be of interest.

5. The authors claim that indications of "chaotic dynamics" are found. To me, Fig.4 is not conclusive. I see no period-doubling, no sudden onset of irregularities, no co-existence of rhythms or other nonlinear phenomena. Nonlinear phenomena can be expected in such a driven nonlinear system but some more evidence should be provided as supplementary material.

6. Why the term "Response fundamental frequency" in Fig.4?

7. "... switching to constant intensity ...". Is there evidence of a Hopf bifurcation? E.g. a square-root dependency of amplitudes?

8. At least the extracted data (areas, time-series...) should be made publicly available since they can provide a rich resource for follow-up studies.

Reviewer #1 (Remarks to the Author):

“This paper focuses on the understudied multinucleate alga of the genus *Caulerpa*, which undergoes morphogenesis despite being a single cell. The paper nicely demonstrates that the alga undergoes circadian rhythms in the localisation of chloroplasts, which migrate to the base of the cell during night. The migratory movements anticipate the arrival of light or dark and can persist under low constant light, demonstrating the action of a circadian oscillator that can be entrained. The paper further shows that growing the alga under a day-night cycle favours growth compared to growing under constant illumination with the same average light intensity, indicating that the circadian response of the organism contributes to its ability to grow and spread. All of these are valuable and novel findings that bring description and analysis of circadian rhythms to this understudied system.”

The report by Reviewer #1 expresses appreciation for the value and novelty of the experimental results and the analysis, as well as for the quality of the paper. We are thankful for the positive report of Reviewer #1.

“However, the link between the presented findings and the mystery of unicellular morphogenesis, which as the authors point out is the most intriguing aspect of *Caulerpa*, is weak. The effect of continuous compared to cyclic illumination seems to be reduced growth, and while this may affect aspect ratio of the frond, this appears to be no more instructive for understanding morphogenesis than showing that the aspect ratio of a leaf can be altered by growing a plant under abnormal conditions that retard growth.”

Others have suggested and discussed an adaptive value of circadian rhythms in plants, for example

Yerushalmi, Shai, and Rachel M. Green. “Evidence for the Adaptive Significance of Circadian Rhythms.” *Ecology Letters* 12, no. 9 (2009): 970–81.

<https://doi.org/10.1111/j.1461-0248.2009.01343.x>

The current manuscript under consideration makes no claims regarding conditions favouring growth. Rather the findings reported in the manuscript indicate a morphological effect of the temporal pattern of illumination, as stated in the concluding phrase of the introduction to the manuscript.

Following the helpful comment by Review #1, we have added new Supplementary material, Supplementary Fig. 5 in the revised manuscript, presenting inferred growth curves for the conditions in Fig. 5. The comparison shows that the morphological effects presented in Fig. 5 are not apparent from the growth curves; see Supplementary Fig. 5. Based on these results we would not make a statement about growth reduction under constant illumination. And yet, one can infer the temporal pattern of illumination from the morphology. This is consistent with the statements in the submitted manuscript.

Points we find insightful, and that are of novelty to the best of our knowledge:

(i) the *abnormal condition* here is the lack of time-dependent environmental light conditions — the absence of distinct night- and day- like intervals.

We start with a reference case, 12hL-12hD at average photon flux of $2^{-1} I_L$ (Fig. 5 lower-left quadrant). Leaving the lights ON, meaning constant photon flux of I_L , leads to shorter fronds (Fig 5. upper-right quadrant). This result is not explained simply by too many photons, as can be learnt from 12hL-12hD at an equivalent average photon flux of I_L (Fig. 5 upper-left quadrant). Moreover, when the average photon flux matches that of the reference, constant $2^{-1} I_L$ (Fig. 5 lower-right quadrant) appears morphologically closer to the constant illumination at I_L .

(ii) the reported results are of a system where the frond tissue- and organ- like regions consist of a continuous single cell.

In addition, the results in the manuscript demonstrate a link between the temporal pattern of light and the waves. These findings suggest the possibility of a link between the waves and morphogenesis. This leads to a new hypothesis presented in the discussion section of the manuscript.

“Moreover, it is unclear whether the effect on growth is caused by the altered migration of chloroplasts or other metabolic consequences of continuous illumination.”

We are working on further investigation of the relationships among the waves, organ morphogenesis, and the metabolism of the organism.

The point by Reviewer #1 lies at the transition between the reported results and their potential implications. We are thankful to Reviewer #1 for pointing this out. To ensure the presentation makes clear distinction between results and potential implications, the revised concluding phrase of the abstract now reads:

“Moreover, we find distinct morphologies depending on light temporal patterns, suggesting waves of chlorophyll could link biological oscillators, metabolism and morphogenesis in this giant single-celled organism.”

The revised caption of Fig. 5 cross-references to the new Supplementary Fig. 5:

“The developmental effect distinguishing between the temporal distributions compared above — T=24h and constant illumination — are not as apparent in growth

curves; corresponding moving averages of the time-series can be found in Supplementary Fig. 5.”

Reviewer #2 (Remarks to the Author):

"This work analyzes the dynamics of chloroplast waves in green algae and how these periodic dynamics entrain to forcing over time. The data and results in this work are quite profound, providing evidence of complex spatiotemporal dynamics and patterns over space and time. These results will have significant implications for the biology and applied math communities. The results are very well presented (excellent figures, very thorough analysis, clear writing). I am strongly in favor of the publication of this work.

One minor revision I would like to see is more detail on the dynamical systems modeling/analysis. Specifically the coupled oscillator model is mentioned, but I had a hard time finding the details in the paper to connect to the figures and results. Even just a quick summary or highlights would be nice. Not necessary though as I realize this is likely a whole set of follow up papers."

We are thrilled to learn of Reviewer #2's excitement and recognition of the broad relevance of the research, approaches and results reported in this paper. Similarly to the Reviewer, we as well are interested in insights that would arise from mathematical modelling of this system, which is beyond the scope of the current manuscript.

To better connect the idea of coupled oscillators to the reported results, while keeping the presentation concise and focused, we have introduced the revisions in what follows.

In the closing paragraph of the introduction the guiding keywords *self-oscillations* and *forced synchronisation* are introduced, as well as a new reference textbook:

Balanov, Alexander, Natalia Janson, Olga Sosnovtseva, and Dmitry Postnov. Synchronization. Springer Series in Synergetics. Berlin, Heidelberg: Springer Berlin Heidelberg, 2009. <https://doi.org/10.1007/978-3-540-72128-4>.

The revised sentences are copied below, and the modifications are underlined:

"... Moreover, the analysis shows that anticipatory behaviour in this system is explained by entrainment — adjustment of the rhythm of an organismal self-sustained oscillator by interaction with another oscillator (Pikovsky, Rosenblum, and Kurths 2001; Balanov et al. 2009), the driving illumination in this case. Using a scalable and affordable methodology, varying two control parameters — the temporal period of the illumination and its intensity — we identify distinct dynamical states of the self-organised waves. The results are compatible with dynamics of self-oscillations and forced synchronisation (Balanov et al. 2009). ..."

The new reference, Balanov et al. 2009, is now included also in the relevant Result section, where n and m are now revised to follow the convention in the cited literature for $n:m$ -entrainment :

"Entrainment of a nonlinear oscillator is predicted to occur at driving frequencies near rational ratios of its natural frequency and the driving one, in which case \underline{n} cycles of the response would match \underline{m} cycles of the driving oscillator.(Pikovsky, Rosenblum, and Kurths 2001; Balanov et al. 2009) ..."

In the same paragraph we have introduced revisions which relate the terms $n:m$ -entrainment , synchronisation regions and Arnold tongues. The cited reference, Herrmann et al. 2020, which has inspired the visualisation in Fig.3c, connects such visualisation with Arnold tongues.

"... Within an $n:m$ -entrainment range, the response fundamental frequency $f_{r,0}$ and its higher harmonics are predicted to be modified accordingly. To test for entrainment, we have measured samples subject to f_d about rational ratios of 24h, namely $T=48h, 6h, 3h,$ and $1.5h,$ corresponding to $2^1, 2^{-1}, 2^{-3}$ and 2^{-4} multiples of 24h, as well as longer ones $T=94h$ and $54h.$ The results support a 1:1-entrainment range, as well as evidence for higher order $n:m$ -entrainment regions. The corresponding power spectra are presented as heatmaps, where the columns correspond to $f_d,$ including those from f_d near $1/24h;$ see Fig. 3c, a visualisation inspired by a study on forced turbulence and its synchronisation regions. (Herrmann et al. 2020) The synchronisation regions, where $n:m$ -entrainment are stable, are known as Arnold tongues. (Pikovsky, Rosenblum, and Kurths 2001; Balanov et al. 2009; Herrmann et al. 2020; Burt et al. 2021)"

Reviewer #3 (Remarks to the Author):

“The authors study an interesting example of spatiotemporal self-organization – the light driven growth of a giant unicellular green alga. Their study combines nonlinear dynamics, developmental biology, and chronobiology. Thus the results should be of interest to a broad range of readers. I find the enormous amount of quantitative imaging data impressive (compare Fig.1). A system was established to combine short term dynamics, daily rhythms, and long-term growth. The huge amount of processed data becomes visible in their comprehensive Figures 3 and 4.

The challenge was to condense all these data to a focused manuscript. Using global characteristics such as normalized areas and Fourier analysis many exciting results could be extracted: indications of self-sustained rhythms, entrainment to a wide range of light-dark cycles, and relevance of entrainment for morphogenesis.

In order to learn even more from these great data of an interesting model organism I list below some technical questions I would like to be discussed. Some answers might be also included into the revised manuscript which is already of high quality.”

We are excited to learn of Reviewer #3’s interest in the research, appreciation of the challenges, approaches and results reported in this paper, and grateful for the Reviewer’s in-depth critical reading and suggestions.

“Specific comments and questions:”

Comment #1

1. “... waves ... propagate over centimetres, within a few hours ...” This propagation seems really fast compared to normal diffusion. The authors might discuss some possible explanations: convection, active transport, anomalous diffusion, trigger waves (see e.g. J Ferrell 2018).

Reviewer #3 raises the interesting question of the mechanism underlying the waves of greenness. As the Reviewer mentions, in this report we apply coarse-graining in space to reduce the spatio-temporal dynamics into a time-series, that of the apparent green area. While the study of the waves in the spatio-temporal domain remains beyond the scope of this manuscript, the following may offer some insight.

Previous studies have reported evidence for active transport of chloroplasts in *Caulerpa*, including actin and microtubules; see Menzel and Menzel (1989) and Refs. therein. This is consistent with our observations under live-imaging microscopy (yet to be published), where cortical chloroplasts are seen moving in adjacent counter moving files, at speeds of a few microns-per-second. Assuming no venation, this is a clear signature of local drive rather than advection by bulk flow.

Thermal diffusion alone would not explain the retracting phase of the wave. Therefore, as Reviewer #3 points out, we do not expect diffusion to provide a complete description of the phenomena.

Reviewer #3 kindly lists trigger waves among the alternatives to diffusion, exemplified in propagating waves of apoptosis through the cytoplasm of *Xenopus laevis* eggs. Cheng and Ferrel (2018) point out that diffusive spread would slow down with increasing distance, while trigger waves would maintain a constant speed and amplitude. Detailed study in *Caulerpa*, underlying the coarse-grained picture, is to be explored in future work.

Menzel, D., and Christine Elsner-Menzel. "Actin-Based Chloroplast Rearrangements in the Cortex of the Giant Coenocytic Green Alga *Caulerpa*." *Protoplasma* 150, no. 1 (February 1989): 1–8. <https://doi.org/10.1007/bf01352915>

Cheng, Xianrui, and James E. Ferrell. "Apoptosis Propagates through the Cytoplasm as Trigger Waves." *Science* 361, no. 6402 (August 10, 2018): 607–12. <https://doi.org/10.1126/science.aah4065>

Comment #2

2. Figure 2 contains a lot of somewhat hidden information. I recommend providing the curves in Fig.2c individually as supplemental material. This could help to address some follow up questions: Is there a kind of resonance (amplitude effects)? Are there the expected relationships between driver period and entrainment phase?

Following the suggestion by Reviewer #3, individual time-series and corresponding power spectra are now included, and are presented in a new Supplementary material, Supplementary Fig. 1 in the revised manuscript. Future work will investigate further resonance and phase relation between the driver and the biological oscillator.

Comments #3 & #7

3. It is claimed in Figure 3 that the intrinsic oscillations are "self-sustained"? Indeed, the rhythms in Fig.3b are persistent over many cycles. It has been shown, however, that a distinction between weakly damped oscillations and limit cycles is difficult even for recordings of more than 40 cycles (TL Leise 2012, D Gonze 2005, PO Westermark 2009). It was shown recently, that light-dark conditions can also drive transitions from self-sustained to damped oscillations (P Burt 2021). The wide range of entrainment (or resonances) in Fig.3c point to a weak intrinsic oscillator that can be entrained easily to a wide range of periods (as in *Neurospora crassa*, see M Merrow 1999).

7. "... switching to constant intensity ...". Is there evidence of a Hopf bifurcation? E.g. a square-root dependency of amplitudes?

Following comment #3 by Reviewer #3, we have replaced "self-sustained" for "nonlinear" in the title of Fig. 3 and in the corresponding main text paragraph of the revised manuscript. The revised choice of terms highlights the distinction from a harmonic oscillator, while not making a statement of its "self-sustained" nature at this stage of the presentation.

Reviewer #3 has proposed insightful literature — Gonze et. al (2005), Westermark et al. (2009), and Leise et. al (2012) — pointing out the possibility of an underlying model of damped oscillators whose oscillations are sustained by noise. Future modelling work may allow favouring this possibility over that of a deterministic limit cycle. Such models and further analysis may allow better separation of phase and amplitude in this system, facilitating the exploration of a Hopf bifurcation.

Relating to the interesting case of a damped harmonic oscillator driven by noise (Westermark et al. 2009): as of now we have not found parameters for which a damped harmonic oscillator driven by noise results in a frequency response compatible with those presented in Fig. 3 of the manuscript.

"It was shown recently, that light-dark conditions can also drive transitions from self-sustained to damped oscillations (P Burt 2021)."

The extensive work by Burt et al. (2021), mentioned in Reviewer #3's comment, combines experiments and mathematical modelling to study entrainment subject to a driving force of alternating temperatures, where the control parameters are the period T and the thermoperiod (duty-cycle) χ_T .

The authors report the range of period and phase entrainment under varying conditions, and explore nonlinear phenomena outside of the 1:1 entrainment range. The authors report experiments conducted under constant illumination (dark), using light for imaging rather than driving the biological system. Damped oscillations are associated with the low temperature phases of the driving protocol, and the authors discuss this in the context of entrainment (Burt et al., 2021).

The data presented in the manuscript under consideration on the waves of greenness in *Caulerpa brachypus* does not support a complete damping out of the oscillations in response to the driving protocol. It would be indeed insightful to identify conditions which lead to waves arrest (a fixed point) while keeping the system alive.

"The wide range of entrainment (or resonances) in Fig.3c point to a weak intrinsic oscillator that can be entrained easily to a wide range of periods (as in *Neurospora crassa*, see M Merrow 1999)."

One way to compare the relative strength of the intrinsic oscillator and the forcing would be experiments with periodic illumination at lower amplitude of the photon flux. These are ongoing experiments and will be reported when completed.

We appreciate the proposed reading by Reviewer #3 — Merrow et al. (1999) — which explores the role of the FRQ gene in the *Neurospora* model system. Fig. 2c in Merrow et al. (1999) reports phase plots based on experiments applying alternating temperature. The experimental protocols include periods within 0.73 and 1.14 of the reported free-running period of the wild-type (22h), as well as down to 0.66 and up to 1.56 of the reported free-running periods of mutants. Also presented in the same plot are data from a strain labelled by the authors *arrhythmic*.

Merrow et al. (1999) state:

"All strains tested, including the FRQ-deficient mutant frq^9 , establish a stable phase relationship to the zeitgeber cycle, although frq^+ and frq^9 are almost 180° out of phase. The two rhythmic strains, frq^1 and frq^7 (FRPs of 16 and 29 h), also entrain in anti-phase (Fig. 2c)",

That is, all strains were reported to show a stable phase at $T=19h$, and down to $T=16h$ for frq^+ (Merrow et al. 1999).

On the other hand, Burt et al. (2021) report that "Conidiation patterns of the frq^+ strain do not entrain to a $T=16h$ thermocycle of $\chi_T=0.5$ "; see caption of Fig. 1 D,E,F, as well as corresponding data in Figs. 2, S2 and S3 (Burt et al., 2021).

Furthermore, Fig. 2 (Burt et al. 2021) shows that frq^7 does not entrain at $T=22h$, $\chi_T=0.5$; see also corresponding data presented in Fig. S6 (Burt et al. 2021).

A resolution of this apparent discrepancy between Merrow et al. (1999) and Burt et al. (2021) merits a venue other than our manuscript, one which allows direct discussion among the authors of the above mentioned publications.

Leise, Tanya L., Connie W. Wang, Paula J. Gitis, and David K. Welsh. "Persistent Cell-Autonomous Circadian Oscillations in Fibroblasts Revealed by Six-Week Single-Cell Imaging of PER2::LUC Bioluminescence." *PLOS ONE* 7, no. 3 (March 29, 2012): e33334.
<https://doi.org/10.1371/journal.pone.0033334>.

Gonze, Didier, Samuel Bernard, Christian Waltermann, Achim Kramer, and Hanspeter Herzel. "Spontaneous Synchronization of Coupled Circadian Oscillators." *Biophysical Journal* 89, no. 1 (July 1, 2005): 120–29. <https://doi.org/10.1529/biophysj.104.058388>.

Westermark, Pål O., David K. Welsh, Hitoshi Okamura, and Hanspeter Herzel. "Quantification of Circadian Rhythms in Single Cells." *PLOS Computational Biology* 5, no. 11 (November 26, 2009): e1000580. <https://doi.org/10.1371/journal.pcbi.1000580>

Burt, Philipp, Saskia Grabe, Cornelia Madeti, Abhishek Upadhyay, Martha Merrow, Till Roenneberg, Hanspeter Herzel, and Christoph Schmal. "Principles Underlying the Complex Dynamics of Temperature Entrainment by a Circadian Clock." *IScience* 24, no. 11 (November 19, 2021). <https://doi.org/10.1016/j.isci.2021.103370>

Merrow, Martha, Michael Brunner, and Till Roenneberg. "Assignment of Circadian Function for the Neurospora Clock Gene Frequency." *Nature* 399, no. 6736 (June 1999): 584–86. <https://doi.org/10.1038/21190>

Comment #4

4. Driving the system with different T-cycles (DJ Dijk 1999) reminds me to the "forced-desynchrony protocols". In these cases no entrainment is expected for the strong human clock. A comparison might be of interest.

Czeisler et al. (1999) and Wyatt et al. (1999) report results from "*forced-desynchrony protocol*" applying T-cycles of 28h and 20h correspondingly: To estimate endogenous circadian period "*core body temperature, plasma melatonin, and plasma cortisol were sampled during the forced desynchrony protocols.*", and were found to be "*highly correlated when analyzed within an individual subject*". Based on earlier reports, the authors stated that the "*28-hour day length on this forced desynchrony protocol was (i) far enough outside the range of entrainment of the human circadian pacemaker so as to minimize the influence of the imposed schedule on the observed circadian period ...*". In fact, the protocol is designed to desynchronize the sleep–wake cycle from other circadian timing processes, and to allow the assessment of their separate contributions to physiology and behavior and to estimate intrinsic circadian period; see Wang et al. (2023).

These reports provide evidence that forced sleep-wake cycles, 4h shorter or longer than 24h, do not result in apparent entrainment of several other circadian processes in the human body, namely core body temperature, plasma melatonin, and plasma cortisol.

In our manuscript we report results from experiments using light, which is the main energy source provided to the organism. As demonstrated in the manuscript, the 1:1-entrainment range of *Caulerpa brachypus* is at least 18-30h, at the light intensity explored as reference level.

While we are exploring methods to entrain the waves of greenness using non photosynthetic means, this remains beyond the scope of the current report. The analogy between the human sleep-wake homeostatic process and the waves in *Caulerpa*, and their coupling to circadian clocks, are certainly thought-provoking.

Czeisler, Charles A., Jeanne F. Duffy, Theresa L. Shanahan, Emery N. Brown, Jude F. Mitchell, David W. Rimmer, Joseph M. Ronda, et al. (including Derk-Jan Dijk) "Stability, Precision, and Near-24-Hour Period of the Human Circadian Pacemaker." *Science* 284, no. 5423 (June 25, 1999): 2177–81. <https://doi.org/10.1126/science.284.5423.2177>

Wyatt, James K., Angela Ritz-De Cecco, Charles A. Czeisler, and Derk-Jan Dijk. "Circadian Temperature and Melatonin Rhythms, Sleep, and Neurobehavioral Function in Humans Living on a 20-h Day." *American Journal of Physiology-Regulatory, Integrative and Comparative Physiology* 277, no. 4 (October 1999): R1152–63. [\https://doi.org/10.1152/ajpregu.1999.277.4.R1152

Wang, Wei, Robin K. Yuan, Jude F. Mitchell, Kirsi-Marja Zitting, Melissa A. St. Hilaire, James K. Wyatt, Frank A. J. L. Scheer, et al. "Desynchronizing the Sleep--Wake Cycle from Circadian Timing to Assess Their Separate Contributions to Physiology and Behaviour and to Estimate Intrinsic Circadian Period." *Nature Protocols* 18, no. 2 (February 2023): 579–603. <https://doi.org/10.1038/s41596-022-00746-y>

Comment #5

5. The authors claim that indications of "chaotic dynamics" are found. To me, Fig.4 is not conclusive. I see no period-doubling, no sudden onset of irregularities, no co-existence of rhythms or other nonlinear phenomena. Nonlinear phenomena can be expected in such a driven nonlinear system but some more evidence should be provided as supplementary material.

Similarly to Reviewer #3, we find that further analysis and modelling is needed to tell apart deterministic low-dimensional chaos from stochastic dynamics, or a combination of these. The manuscript states "*aperiodic dynamical states*", "... the dynamics exhibits a transition to *temporal disorder*", and "a manifestation of the nonlinear nature of the dynamics, and a *plausible signature of chaotic dynamics*".

Subject to the higher constant photon flux, the time series in Fig. 4b appear to transition intermittently between regular oscillations and other excursions. This is also reflected by the increase in power distribution across a continuum of frequencies. Attractor reconstruction and Lyapunov exponents analysis will be part of future work. In the meantime we have

added the time series and corresponding power spectra for the samples in Fig. 4. These are presented in new supplementary material, Supplementary Fig. 4 in the revised manuscript.

We are thankful to Reviewer #3 for pointing out that using the term "chaotic dynamics" may beg for further elaboration. To clarify, a related discussion sentence in the revised manuscript now reads:

"The latter leads to increased temporal disorder dynamics, hinting that relaxation in the dark is essential for the typical organismal dynamics."

Comment #6

6. Why the term "Response fundamental frequency" in Fig.4?

Using the term *response fundamental frequency* we follow the convention that has been set earlier in this manuscript. The terms *response frequency* f_r and *response fundamental frequency* $f_{r,0}$ appear in Fig.2b–e and caption, Fig.3c, as well as in the main text, for example:

"To quantify the temporal frequency content of the biological response, we apply power spectral analysis." (opening of Sec. "Anticipatory green waves and morphogenesis")

Specifically Fig.4c shows the fundamental frequencies inferred from the power spectra of samples regenerating under constant illumination; these power spectra are presented in new supplementary material, Supplementary Fig. 4 of the revised manuscript, which averages are plotted in Fig.4d. Correspondingly in the main text it is stated:

"(i) the response fundamental frequency increases with decreasing illumination intensity, see Fig. 4c;"

The modifier *response* highlights that these fundamental frequencies are associated with the organismal output or behaviour in response to the *driving* illumination, namely frequency f_d and intensity I_d . Using the term *response* we follow a common convention in the literature, for example:

Balanov, Alexander, Natalia Janson, Olga Sosnovtseva, and Dmitry Postnov. *Synchronization*. Springer Series in Synergetics. Berlin, Heidelberg: Springer Berlin Heidelberg, 2009. <https://doi.org/10.1007/978-3-540-72128-4>

Herrmann, Benjamín, Philipp Oswald, Richard Semaan, and Steven L. Brunton. "Modeling Synchronization in Forced Turbulent Oscillator Flows." *Communications Physics* 3, no. 1 (October 30, 2020): 1–9. <https://doi.org/10.1038/s42005-020-00466-3>

While this may not be the point of comment #6 by Reviewer #3, we realise that the concept of *response* may not be clear in the context of *free-running*. Free-running period, as commonly used in the chronobiology literature, indicates constant conditions; for example

“Period of a rhythm free running in constant light (dark) and constant temperature.”

Pittendrigh, Colin S. “Circadian Rhythms and the Circadian Organization of Living Systems.” *Cold Spring Harbor Symposia on Quantitative Biology* 25 (January 1, 1960): 159–84. <https://doi.org/10.1101/SQB.1960.025.01.015>

In the manuscript under consideration, *free-running* is meant to highlight constant-in-time conditions, as opposed to forced oscillations by a periodic driving force. Yet, the system is still being driven, and exhibits a response fundamental frequency varying with the intensity of the illumination. Hence *free-running* appears in the current manuscript only when it comes with its meaning.

The following revisions have been introduced for clarity, edits are underlined:

Sec. “Equivalence of dynamics near 24h driving periods”:

“The pronounced local maxima in the power spectra correspond to the response fundamental frequency $f_{r,0}$, centred at 1/24h for these samples, and its higher harmonics — integer multiples of $f_{r,0}$.”

Caption of Fig. 4c

“Response fundamental frequencies $f_{r,0}$ plotted as function of the driving intensity, I_d ; the data indicate an intensity dependence of the response fundamental frequency when the conditions are constant-in-time.”

And related main text:

“(i) the response fundamental frequency $f_{r,0}$ increases with decreasing illumination intensity, see...”

Comment #7

7. “... switching to constant intensity ...”. Is there evidence of a Hopf bifurcation? E.g. a square-root dependency of amplitudes?

See reply to combined comments comments #3 and #7 by Reviewer #3.

Comment #8

8. At least the extracted data (areas, time-series...) should be made publicly available since they can provide a rich resource for follow-up studies.

We are excited that the Reviewer recognizes the value of the datasets for the community. Data Availability has been revised with links to the dataset.

REVIEWERS' COMMENTS

Reviewer #1 (Remarks to the Author):

I thank the authors for responding to my comments. My only concern was the link between chlorophyll waves and morphogenesis is weak. In their response, the authors clarify that their findings indicate a morphological effect of the temporal pattern of illumination, as stated in the concluding phrase of the introduction. That is, they demonstrate an effect of the illumination pattern on morphology not of chlorophyll waves. It could be that the illumination effect operates through the chlorophyll waves or via some other route, in parallel with the generation of chlorophyll waves. Their results do not discriminate between these possibilities. The authors have rightly modified their abstract by having "could link" instead of "link" in their final sentence about the connection between morphogenesis and waves. However, the title still states: "Macroscopic waves link biological clocks, light and morphogenesis in a giant unicellular green alga", which indicates that a link between waves and morphogenesis has been demonstrated.

I understand the authors' enthusiasm to link their findings to morphogenesis. As the authors state in their introduction: "Caulerpa presents the mystery of morphogenesis on macroscopic scales in the absence of cellularization." However, this enthusiasm should not lead to them make misleading claims in their title. I would therefore encourage the authors to modify their title so that it accurately reflects what their paper shows.

Reviewer #2 (Remarks to the Author):

I'd like to thank the authors for their careful revision and for addressing all of my concerns. I am happy to recommend publication of this very strong and interesting paper.

Reviewer #3 (Remarks to the Author):

Thanks for the careful discussion of the issues raised in the reviews. I appreciate the addition of the excellent reference to the Balanov-book - a perfect introduction to oscillator theory. Furthermore, I find the improvements of the terminology helpful. Now the authors are appropriately conservative in their claims. I appreciate also the addition of comprehensive Supplementary Material and raw data as a starting point of follow-up studies.

NCOMMS-23-16182A — point-by-point response

Reviewer #1 (Remarks to the Author):

“I thank the authors for responding to my comments. My only concern was the link between chlorophyll waves and morphogenesis is weak. In their response, the authors clarify that their findings indicate a morphological effect of the temporal pattern of illumination, as stated in the concluding phrase of the introduction. That is, they demonstrate an effect of the illumination pattern on morphology not of chlorophyll waves. It could be that the illumination effect operates through the chlorophyll waves or via some other route, in parallel with the generation of chlorophyll waves. Their results do not discriminate between these possibilities. The authors have rightly modified their abstract by having "could link" instead of "link" in their final sentence about the connection between morphogenesis and waves. However, the title still states: "Macroscopic waves link biological clocks, light and morphogenesis in a giant unicellular green alga", which indicates that a link between waves and morphogenesis has been demonstrated.

I understand the authors' enthusiasm to link their findings to morphogenesis. As the authors state in their introduction: "Caulerpa presents the mystery of morphogenesis on macroscopic scales in the absence of cellularization." However, this enthusiasm should not lead to them make misleading claims in their title. I would therefore encourage the authors to modify their title so that it accurately reflects what their paper shows.”

Reviewer #1 raises a concern that the previously submitted title may be misleading with regards to the findings reported in this manuscript. Considering such undesired reading, and to emphasise relations which have been established in this work, we have revised the title.

We thank all three Reviewers for their valuable comments, for helping us improve the presentation of this work, and for recognizing its value and novelty.

Reviewer #2 (Remarks to the Author):

“I'd like to thank the authors for their careful revision and for addressing all of my concerns. I am happy to recommend publication of this very strong and interesting paper.”

Reviewer #3 (Remarks to the Author):

“Thanks for the careful discussion of the issues raised in the reviews. I appreciate the addition of the excellent reference to the Balanov-book - a perfect introduction to oscillator theory. Furthermore, I find the improvements of the terminology helpful. Now the authors are appropriately conservative in their claims. I appreciate also the addition of comprehensive Supplementary Material and raw data as a starting point of follow-up studies.”